# NLA-Bit: A Basic Structure for Storing Big Data with Complexity O(1)

**Krasimira Borislavova Ivanova** 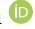

Information Technology Department, University of Telecommunications and Post, 1700 Sofia, Bulgaria; krasy78@mail.bg

**Abstract:** This paper introduces a novel approach for storing Resource Description Framework (RDF) data based on the possibilities of Natural Language Addressing (NLA) and on a special NLA basic structure for storing Big Data, called "NLA-bit", which is aimed to support middle-size or large distributed RDF triple or quadruple stores with time complexity O(1). The main idea of NLA is to use letter codes as coordinates (addresses) for data storing. This avoids indexing and provides high-speed direct access to the data with time complexity O(1). NLA-bit is a structured set of all RDF instances with the same "Subject". An example based on a document system, where every document is stored as NLA-bit, which contains all data connected to it by metadata links, is discussed. The NLA-bits open up a wide field for research and practical implementations in the field of large databases with dynamic semi-structured data (Big Data). Important advantages of the approach are as follow: (1) The reduction of the amount of occupied memory due to the complete absence of additional indexes, absolute addresses, pointers, and additional files; (2) reduction of processing time due to the complete lack of demand—the data are stored/extracted to/from a direct address.

**Keywords:** Big Data; Natural Language Addressing; NLA-bit; DBMS time complexity O(1)

## 1. Introduction

Traditional databases are built by taking into account the presence of regularity or homogeneity in the data. Regularity is a principle in design standardization, providing an abstract view of the world in which exceptions to the rules are not taken into account, insofar as they are considered insignificant when designing a well-structured scheme. At the same time, the a priori homogeneity required by the relational model, as a rule, leads to a lack of flexibility in modeling the dynamics, as found in semantic web data [1].

The variety of data forms and types has led to the introduction of a "semi-structured data model". This model better fits web data such as HTML and XML [2]. As a rule, basic approaches to storing semi-structured data either map the data to relational databases, or use non-relational databases and/or flat files and indexes [3].

Semi-structured data have emerged as a leading topic, as there are a number of data sources that cannot be limited by a schema, but need to be organized in some kind of database. In addition, the need for an extremely flexible data-exchange format between different databases is becoming increasingly important [4].

### 1.1. Graph Databases

Many of the semi-structured models are "graph-like". This brought attention back to the graph models. These are models in which the data structures for the schema and/or instances are represented as a directed, probably labeled, graph. Data manipulation is expressed through graph-oriented operations [5].

Database graph models emerged in the 1980s and early 1990s, and their influence gradually diminished with the advent of other database models, in particular geographic, spatial, semi-structured and XML models. Graph databases are used in areas where

information about the interconnectedness or topology of the data is more important than, or as important as, the data themselves. Graph databases use a model in which the data structures for the schema and/or instances are represented as an oriented graph, and the data manipulation is set by graph-oriented operations [5].

We may classify the graph databases' models in two main classes:

- With explicit schema, such as Gram [6], GMOD [7], PaMaL [8], GOOD [8,9], GOAL [10], GDM [11,12], Logical Data Model (LDM) [13,14], Hypernode Mode (HyM) [15–17], and GROOVY [18]. The explicit schema exists separately from the data in the database and is used as an external pattern for data organization.
- With implicit schema such as RDF [19–21], GGL [22–25], Simatic-XT [26], and Object Exchange Model (OEM) [27]. Implicit schema is integrated in the database as a root of the data structure. In other words, the first data structure which can be accessed is the schema from which all other structures may be traversed, using it as a pattern.

### 1.2. Resource Description Framework

The W3C's recommendation, called "Resource Description Framework" (RDF), is a standard syntax for semantic web annotations and languages [19,28]. There is a growing amount of data that is not structured enough to be supported by traditional databases, but contains regular structures that can be used in queries. Thus, the main purpose of RDF is to describe and allow the processing of irregular or semi-structured data [29].

The basic structure of RDF is a collection of triples, each of which consists of a Subject, a Relation and an Object. The set of RDF triples is called an RDF graph. The RDF standard allows RDF quadruple structure which consists of Subject, Relation, Object and Context [30]. We assume that the triple is a quadruple with an empty Context.

An important advantage of RDF as a storage language is the ability to connect different data sources through a number of additional triplets, to represent the new connections. With the Relational Database Management Systems (RDBMS), this is very difficult because the schema may need to be redesigned. In addition, RDF provides great flexibility as a result of the diversity of the graph-based model. In other words, almost any data type can be described by RDF triplets, with no limit to the size of the graph, whereas, in RDBMS, the scheme must be short. This is important when the data structure is not known in advance. Finally, different types of knowledge can be presented through RDF. This allows the extraction and reuse of knowledge from different sources [31].

Therefore, RDF offers a useful data format, which requires effective management. This is a serious problem when working with a huge number of RDF instances. RDF repositories should provide at least the basic operations on RDF data: searching, updating, inserting and deleting RDF triplets.

In practice, different approaches are used to store RDF data: in main memory, in files or in databases.

For small applications and for experimental purposes, storing and processing RDF schemas and instances in RAM are useful because they can be effectively accessed and manipulated. However, this cannot be a serious method for storing and processing dynamic and very large volumes of data [32].

In order to enable permanent data storage, means for access to indexed files have been developed, but with the increase in the amount of data, it is imperative to use a database management system. Relational and Object-relational database management systems are mainly used.

In general, the systems for RDF data management are based on the following (see also Reference [33]):

- Structures in memory (TRIPLE [34], BitMat [35], Hexastore [36], Jena [37,38], YARS [39], and Sesame [40,41]);
- Popular relational databases (ICS-FORTH RDF Suite [42,43], Semantics Platform 2.0 of Intellidimension Inc., Belfast, ME, USA [44], Ontopia Knowledge Suite [45],

Hexastore [36], Jena [37,38], 3store [46], 4store [47], Kowari [48], Oracle [49], RDF-3X [50], RDFSuite [42], Virtuoso [51], and Sesame [40,41]);

- File systems, usually indexed by key B-trees and/or non-relational databases, such as Oracle Berkeley DB (Sesame [40,41], rdfDB [52], RDF Store [53], Redland [54], Jena [55], Parliament [56], and RDFCube [57]).

### 1.3. Technologies for Storing RDF Data

Good reviews of existing technologies for storing and retrieving RDF data are given in References [1,58]. Technologies for storing RDF data can be classified into the following:

(1) General schemes, i.e., schemes that do not support certain structures and run on third-party databases, such as Jena SDB, which can connect to almost all relational databases, such as MySQL, PostsgreSQL and Oracle;

(2) Specific to schemas that provide storing with their own database structures (Virtuoso, Mulgara, AllegroGraph and Garlik JXT).

#### 1.3.1. General Schemes

Vertical Presentation

This is the simplest RDF schema with only one table needed in the database.

The table contains three columns, called Subject, Relation and Object. Indexes are added for each column [59]. The biggest advantage of this scheme is that it does not require restructuring in case of data changes. This approach is used by 3store [46], RDFStore [53], Redland [54], Oracle [60,61], and rdfDB [62]. In many publications "Relation" is named as a "Property" with the same meaning.

Normalized Triple Store

An additional idea is the normalized triple store. It consists of adding two additional tables to store unique registration identifiers (URIs) and literals separately, which requires significantly less storage space [46] (Table 1). A hybrid of simple and normalized triple storage is also possible, in which the storing of the values themselves is either in the triple table or in the resource table [37].

**Table 1.** Normalized triple store.

| Triples: | | | | Resources: | | Literals: | |
|---|---|---|---|---|---|---|---|
| Subject | Relation | Is Literal | Object | ID | URI | ID | Value |
| r1 | r2 | False | r3 | r1 | … #1 | l1 | Value1 |
| r1 | r4 | True | l1 | r2 | … #2 | … | … |
| … | … | … | … | … | … | … | … |

URI, unique registration identifier.

The table can be divided horizontally into several tables, each of which represents a separate Relation. These tables only need two columns for Subject and Object. Table names implicitly contain Relations [63,64].

To implement the vertical partitioning approach, the tables must be stored by using a column-oriented Database Management System (DBMS) (i.e., a DBMS designed specifically for this case). The columns can be indexed (e.g., using a non-clustered B + tree). The main advantage of vertical partitioning is the support for fast object join. This is achieved by sorting tables. The vertical partition approach offers support for multivalued attributes. In fact, if an object has more than one object value for a given property, each individual value is listed sequentially in the table for this property. For a query, only the Relations included in that query need to be read, and no clustering algorithm is required to split the table of triplets into tables with two columns. Inserts are slow because multiple tables need to be accessed [1].

### 1.3.2. Specific Schemas

Horizontal Presentation

The basic schema consists of a table with a column for the instance identifier (ID), a column for the class name and a column for each Relation. Thus, one row in the table corresponds to one instance. This scheme corresponds to the horizontal presentation [59] and obviously has several drawbacks: a large number of columns, high sparsity, an inability to handle multi-valued properties, the need to add columns to the table when adding new properties to the ontology, etc.

Vertical Decomposition

The vertical decomposition results in one Relation table with only two columns for Subject and Object. It is called a decomposition storage model. Relationships are also stored in tables, e.g., the rdf:type table contains the relationships between instances and their classes [58]. A similar hybrid scheme is used to use the combination of both class table and class table Relation schemes (Table 2) [65].

**Table 2.** RDF hybrid schema (the table-per-relation approach).

| ClassA: | Relation1: | | ClassB: |
| --- | --- | --- | --- |
| ID | Subject | Object | ID |
| . . . #1 | . . . #1 | . . . #3 | . . . #3 |
| . . . | . . . | . . . | . . . |

The main disadvantage of this approach is the generation of many NULL values, as not all Relations will be defined for all Subjects. In addition, the ambiguous attributes that are common in RDF data are difficult to express. When searching for all defined Relations of an object, the scanning of all tables is required. The inclusion of new relationships also requires the addition of new tables [1]. This approach has been used by systems such as Jena2 [37,38], Sesame [40,66], RDFSuite [42], and 4store [47].

Multi-Indexing

The idea of multi-indexing is based on the fact that relationship-related queries are not necessarily the most interesting or popular type of queries found in the real-world semantic web applications.

RDF data should be processed equally by the following types of requests:

- Triples with the same Subject;
- Triples with equal Relations;
- List of Subjects or Relations related to an Object.

To this end, a set of six indexes is maintained, covering all possible access schemes that an RDF request may require. These indices are RSO, ROS, SRO, SOR, ORS and OSR (R means Relation, O for Object and S for Subject).

At first glance, such multiple indexing would lead to a combinatorial explosion for a simple relational table. Nevertheless, this is quite practical in the case of RDF data [30,36]. The approach pays equal attention to all elements of RDF [1]. This approach has been used by tools such as the BitMat [35], Hexastore [36], Kowari system [48], RDF-3X [50], Virtuoso [51], Parliament [56], RDFCube [57], TripleT [67], BRAHMS [68], RDFJoin [69], RDFKB [70], and iStore [71].

*1.4. The Goal of This Paper*

The analysis of the considered tools showed that they all use one or more indexes for access to the data. Thus, their complexity is at least O(log n).

In this paper, we introduce a novel approach for storing RDF data based on the possibilities of Natural Language Addressing (NLA) [72,73] and on a special NLA basic structure for storing Big Data, called "NLA-bit", which is aimed to support very large distributed RDF triple or quadruple stores with time complexity O(1).

In all traditional relational databases, continuous reconstructions of the index structures must be done due to the incoming dynamic data. This is a major and extremely serious problem. The volume of data that is collected and indexed continuously soon becomes so large that traditional databases become overloaded and their work becomes extremely slow as they begin to use almost all the time for their own self-maintenance.

NLA-bit does not require such updates. This prevents overloading and slowing down the operation of databases, even with the accumulation of huge arrays of data. The speed of work is constant and independent of the volume of data, i.e., there is a constant time complexity O(1).

In addition, important advantages of the approach are as follows:

- The reduction of the amount of occupied memory due to the complete absence of additional indexes, absolute addresses, pointers and additional files;
- Reduction of processing time due to the complete lack of demand—the data are stored/extracted to/from a direct address.

The NLA-bit is a fundamentally new structure to database organization that does not replace, but naturally complements, other widely used structures of database management systems.

By applying the idea of NLA to the dynamical perfect hash tables supported by MDNDB™ (Multi-Domain Numbered Data Base™) [74], it is possible to realize the same approach as multi-indexing without indexes but with direct access to all data elements. In addition, because of not storing keywords used as addresses, the solution will occupy les memory.

Thus, the structure NLA-bit introduced in this paper may be used in three variants: for Subjects, for Relations and for Objects. Below we outline only the case for Subjects.

Presented in this paper is research that continues the work on possibilities of Natural Language Addressing presented in 2015, in Reference [72], to establish special structures suitable for working just with RDF instances. It is important that the instances may be triples, quadruples or with many more elements. Proving the possibilities of dynamical perfect hash tables for NLA was done in Reference [72]. In this work, the research continues with defining a new structure which may be useful for building RDF stores.

*1.5. Organization of the Paper*

The paper is organized as follows. The Section 2 introduces the main concepts (Natural Language Addressing and NLA-bit). Section 3 outlines an example based on a document system, where every document is stored as NLA-bit which contains all data connected to it by metadata. Section 4 presents discussion of the main results. The paper is finished by a conclusion and the future-work section, two Appendixes, and References.

## 2. NLA-Bit

*2.1. Natural Language Addressing (NLA)*

Natural Language Addressing (NLA) consists of using codes of the letters as coordinates (addresses) for storing the data. This avoids indexing and provides high-speed direct access to the data with time complexity O(1). For example, let us have the following document:

| document identifier: | A519701/2 |
| document data: | letter of conformance of agreement for collaboration |

In computer memory, this can be stored at a file at location address "00971555", and the index pair "key + pointer" is ("A519701/2", "00971555").

The main text of the document (data) is stored, starting from the address "00971555". To read it, we first have to find the document identifier (name "A519701/2") in the indexes and then to access the address "00971555" in the file, to extract the text of the document.

In the same time, the name "A519701/2" is encoded by nine numbers (letters) if we use the ASCII codes. "A519701/2" will look like this, (65, 53, 49, 57, 55, 48, 49, 47, 50), and we can use this vector of codes as co-ordinates for direct addressing in a multidimensional information space (file).

## 2.2. Advantages of NLA

To realize Natural Language Addressing, we use dynamic perfect hash tables [72].

Hash tables are attractive because of the constant algorithmic complexity they can achieve. However, collisions can lead to a significant increase in execution time. Because of this, we cannot use name encoding as hash table keys. To resolve this problem, a special type of file internal organization with special additional indexing was established. Using dynamic perfect hashing is good for the following reasons:

1. The function that uses the encoding of letters with integers locates unambiguously and there is no way to get collisions;
2. This function can be used recursively for each string character and to build perfect hash tables on many levels and thus to have quick access to the data.

For example, the array "65, 53, 49, 57, 55, 48, 49, 47, 50" can be considered as a route to a point in a multidimensional information space and the text of the document can be stored at this point. The hashing function is recursive and builds a hierarchical multilayer set of tables. In the case of A519701/2, we have nine levels.

Document identifiers may be arbitrary long. The length of the words is different, too, and it is possible to use phrases. The set of all natural words and phrases defines a multidimensional space with different dimensions and unlimited size. It is needed a special algorithm to convert these (logical) addresses in (physical) addresses on the hard disk and a program to implement the algorithm, and which would allow the creation of a new type of document-storing system.

This is realized in the MDNDB™ (Multi-Domain Numbered Data Base™) tool system. MDNDB™ is a multi-model database which works with distributed multi-space databases, in particular with RDF graphs. MDNDB™ is based on the access methods ArM32™, BigArM™ and NL-ArM™ and is the instrumental tool for the "INFOS™" system. The MDNDB™ has been used in many practical solutions since 1990 [74].

## 2.3. NLA-Bit

Let us remember that the RDF triple is a statement of a relationship between the following: "Subject"; "Relation" (also called a predicate or property), which denotes a relationship; and "Object", which may be raw text. RDF provides a general method to decompose any information into pieces called triples. RDF quadruple is an RDF triple with additional fourth element "Context", which may be raw text, too [75]. For our research, if the "Context" is empty, then the quadruple is assumed as a triple.

**Definition 1.** *The structured set of all RDF instances with the same "Subject" is called "NLA-bit".*

NLA-bit is structured due to different "Relations" with corresponded "Objects". The "Subject" may have the same Relation with many different "Objects". This way the "Relation" connects the "Subject" with a set of objects, which, in particular, may be empty.

The set of "Objects" connected by a same "Relation" to the same "Subject" is called NLA-bit Layer.

It is assumed, by definition, that every NLA-bit has an unlimited number of layers, as well as that every layer has an unlimited number of objects.

If the practical solution needs processing of requests based on NLA-bits for Relations or NLA-bits for objects, they may be used in parallel with NLA-bits for Subjects.

If the Subject is connected with multiple Relations, they are separate addresses in the NLA-bit space and no conflicts exist.

If a Subject with concrete Relation is connected to multiple Objects, i.e., to the set or multi-set of Objects, the realization solution may be trivial to apply classical concatenation of all Objects in one address point, or specific for NLA to use additional RDF elements such as Context to address separated different values of the Objects.

For the last case, a new "composite" Object is created by composing "Object" and "Relation", i.e., <object: relation>, and a new RDF triple is created by using the "Context" as relation and "Object" as itself. The RDF triple is as follow: **<object:relation> <context> <object>** and corresponded NLA-bit will be for the composed new "Subject", i.e., for (**object:relation**). This approach may be used recursively if it is needed.

Let remark that the multi-sets may be converted in usual sets with unique elements by numbering. In our case, if the "Object" is a multi-set than corresponded numbers may be added to the "Context".

If the "Context" is empty, the receiving time of the triple is used as "Context" and no numbering is need.

The complexity of this, specific for NLA-bits approach, is again O(1).

Interesting case is when the Subject and Object are connected by multiple Relations. One possible realization of such a case is to use NLA-bits for relations. Another variant is to use separate points for all combinations of Subject and Relations which will store the Object many times in all corresponded points. At the end, the most efficient variant is to use Subject and Object as NLA addresses and to store all relations in corresponded points, using additional NLA addresses. In all variants, the complexity is O(1).

The physical organization of NLA-bits is by multi-dimensional information spaces [72]. In Figure 1, a two-dimensional variant is illustrated. Vertical arrows represent the NLA-bits, and horizontal planes represent the layers (relations). NLA-bits with same relation are just points in the same relation's plane. It is assumed that all relations' planes exist, but more of them are empty. This means that if any NLA-bit does not have a given relation, then it is assumed that it has this relation with empty <Object>. This organization supports complexity O(1).

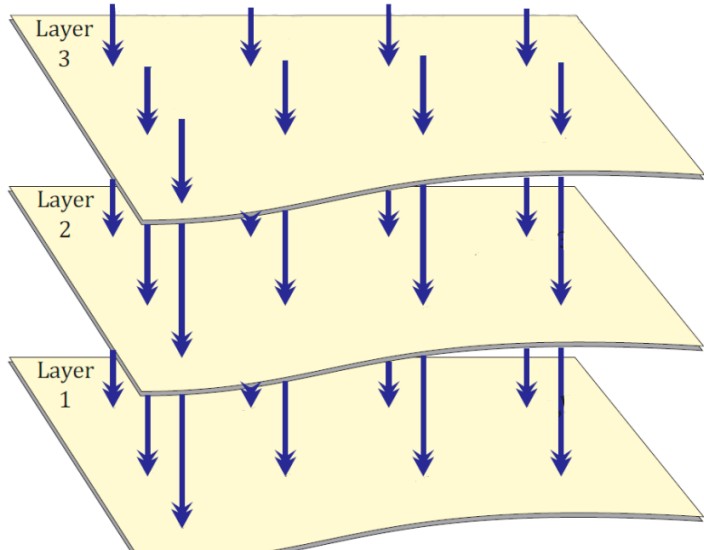

**Figure 1.** Natural Language Addressing (NLA)-bits with same relations (layers).

An example of the NLA-bit for the "Subject" given by name of document "A519701/2" with layers of its metadata is presented in Appendix A Table A1. Only data written in bold

are stored in the archive. Similar example for NLA-bit "CUT" of "Word Net Thesaurus" [76] is given in Reference [72] (pp. 143–147).

The name of the document and names of the relations (layers) are NLA-addresses. The RDF triples which correspond to NLA-bit for the document A519701/2 are shown in Appendix A Table A2.

For relation <read>, there is a multi-set of <objects> and a corresponded NLA-bit <A519701/2: read> is created (Appendix A Tables A3 and A4).

## 3. Results

As an example of the present work, we focus on the administrative documents and the document turnover in the administrative state structures, insofar as there are a number of regulatory laws and regulations. Since the documentary processes in non-governmental organizations are analogous, the proposed approach here could be applied to them without much change.

### *3.1. Use Case: The Document Flow*

3.1.1. Administrative Document and Document Flow

The term "document" has Latin origins (*documentum*) and means proof or testimony. "Administrative document" is a material Object containing data, whose main task is to store in time and space certain official data. The term "document flow" includes all interrelated processes regarding the movement of administrative documents, from the moment of their compilation or receipt to the moment of their final processing, sending or postponing for storing in an archive. The central and territorial administrations process documents of different volumes.

The types of documents—internal, external and outgoing—are defined as such in their specifics by the regulations, competencies and functions of the institutions.

Internal documents of the institution are orders, decisions, instructions and regulations; reports, statements, opinions, inquiries and information; draft regulations; plans, schedules and protocols; letters between the structural units of the institution; applications of employees; business notes, etc.

Outgoing documents of the institution are reports, statements, information and inquiries; drafts of normative documents, initiative letters, answers to incoming documents, cover letters, reminders and additional letters to initiative documents, etc.

Incoming documents are the letters arriving at the institution, accompanied or not by additional documents/annexes.

The documents received or created in the institutions are accepted in the office/registry, where a set of activities determine the way of processing, movement, use of the documents and their archiving.

The most common document systems are organizational and administrative documentation; financial and accounting documentation; commercial documentation; and scientific and technical documentation.

The state administration mainly works with general administrative documents, which can be divided into two main groups:

1. Organizational and administrative documents—order, decision and decree; reference, etc.;
2. Information documents—report, memorandum, information, official letter, report, protocol, list, reference, etc.

"Electronic document" means any content stored in electronic form, in particular text or audio, visual or audio–visual recording.

Electronic documents can be structured or unstructured and must use open formats. Electronic documents with structured content are electronic documents that have a predefined structure through a generally accepted standard. For example, ISO/IEC 26300-OASIS Open Document Format for Office Application is used for text documents, spreadsheets and presentation documents, unless there is a justified technological need for another

format. Electronic documents with unstructured content are all other electronic documents [77]. All received and sent electronic documents are stored in the information system of each administration.

### 3.1.2. Administrative Information System

The heads of the administrations ensure the development and implementation of an Administrative Information System (AIS) in the administrations headed by them. Administrative Information Systems ensure the maintenance and processing of data on the turnover of electronic documents and paper documents in the provision of administrative services and the implementation of administrative procedures. Procedures are all work processes in the administration or between the different administrations, including the internal turnover of documents, but they do not constitute the provision of administrative services and internal electronic administrative services. The AIS maintains and ensures the storing of the received and created electronic documents for a period of no less than 20 years, in a way that allows the reproduction of the documents without data loss.

### 3.1.3. Information Objects

The Administrative Information System maintains a set of related data—information Objects, on which as inseparable units are applicable functions for creation, destruction, access management and other functions. The data in the composition of the information Object are created automatically or manually in the AIS.

In order to service the joint maintenance of electronic documents and paper documents, information Objects of the "document" type are maintained in the AIS. The file content of the electronic documents is one of the data in the composition of an Object of the type "document" and is stored only with the means of AIS. The control of the access to the file content of the electronic documents is performed only with the means of AIS. To access it, a connection is maintained between the description of the information Object with which the electronic document is presented and its file content.

If other data are received together with the file content in the AIS, their processing is performed in connection with the manner of receiving the document in the AIS and in accordance with the requirements of the ordinance. The names of the documents in the AIS are formed according to the internal rules of the administration. The names of the documents can be formed automatically when they are created in the AIS on data submitted by external persons, according to a pre-established scheme in an automatically executed algorithm or in another way.

### 3.1.4. Electronic Correspondence

The correspondence is a set of thematically related documents. An electronic correspondence is created in the AIS on a document requesting an electronic administrative service, including when the document itself is on paper, but a desire to provide the service electronically or a processing procedure is explicitly stated. Each of the sections is presented with a list of links to documents in the AIS, classified in the respective section. The Administrative Information System shall ensure the inclusion of the same document in the relevant sections of any number of correspondences or the inclusion of the same correspondence in the internal sections of any number of other correspondences.

### 3.1.5. Metadata for Documents

The unique identifiers for electronic documents must be used within the AIS. They are accompanied by metadata, providing information for each individual record in the database and revealing all actions with it from the moment of its creation and throughout the life cycle of the document.

Metadata are used to describe the document and reproduce the links between the document and the activities around it, as well as the relationship of individual electronic documents within the entire document system. For each e-document in the electronic

archive, it is necessary to maintain a number of metadata, such as unique registration identifier (URI); type—an explanation of the type of document, ensuring its unambiguous identification by entering the relevant text; concerning—a brief presentation of the essence through text; data on the authorized person, who is authorized to sign the respective type of document; correspondent-name/title; address; PIN/UIC, etc. Data on the storing of the document from the departmental nomenclature presents the term and the schemes for storing of the documents, which manage their stay in the AIS until the transfer of the document to the National Archive Fund; the type of file format for the respective document; attached files' list (name, content); additional comment; additional data; deadline for completion of work on the document; employee, the attention to which it should be directed; e-signature of the document, including the signatures of the employees who prepared and agreed on the document; recipient registration number; method of sending/receiving; creating software; volume of the archive file and the file with the content of the document and its appendices; case index from the nomenclature of cases; chronology of the activities related to the storing and audit of the document; etc.

In addition, there are obligatory requisites to which the electronic documents must correspond. For example, such details are as follows: "Address of the sending organization"; "Address of the recipient"; "About"; "On your №... "; "To №... "; "Address"; "Application"; "Compiled" and "Agreed"; "Signature"; "Resolution"; "Note"; "Transcript"; etc. These details also in a certain way identify the documents and are a type of metadata, which are set in the document in a form implicit for AIS. With the means of artificial intelligence for content analysis, these details, as well as many other data present in the document, can be turned into metadata for the document, and, through it, they can be used to access its content.

### 3.1.6. Example of a System from Practice: DocuWare System

DocuWare is a complete system for digital management of information and processes [78]. DocuWare is a business information and process management software solution that has won the trust of over 18,000 companies and over 500,000 users worldwide.

The DocuWare system allows users to optimize all stages of document processing, access and retrieval of information, and it facilitates collaboration and efficient workflow.

The documents are easily accessible to all authorized users, allowing them to work together on shared files, add notes and annotations, comments, mark individual places in the text and put electronic stamps.

DocuWare allows the creation of clearly structured information management processes. In this way, the work is systematized: The processing of documents is performed electronically and automatically follows a predetermined sequence. Authorized users have the ability to track the movement of the document and the progress of each task.

### *3.2. Experimental System Design*
### 3.2.1. The NLA_Doc System

When designing and developing automated systems for servicing document management activities, the administrations need to follow the legal and regulatory requirements governing the main processes of document management—collection and storing of documents, extraction of essential data, processing and presentation of detailed or summarized user results, etc. As a rule, all of these requirements are already covered by the existing AIS and should not, with implemented and well-functioning AIS, go to the design and implementation of new ones. The correct approach is to expand the capabilities of existing systems with new functionalities that were not available at earlier stages.

This is the leading idea of the current work. By preserving all available functionalities for data storing for the document circulation of the existing AIS, we can offer an extension that would allow a qualitatively new type of organization of data storing for documents that would supplement the already realized possibilities in the available databases.

Here we consider a supplement distributed system for storing document-flow-data based on NLA-bits called NLA_Doc.

### 3.2.2. NLA_Doc Data Structures

The main idea for storing in NLA_Doc is through RDF triplets of the following type:

$$\textbf{<document ID> <metadata ID> <data>} \tag{1}$$

where all three elements are natural language strings.

All data are organized in NLA-bits. Every document is stored as NLA-bit, which contains all data connected to it by metadata.

Metadata identifiers are accepted as NLA-bit's layers, and document identifiers are accepted as names of NLA-bits. Both <document ID> and <metadata ID> are not recorded in the archives—they are natural language addresses. Only <data> are stored in appropriate containers located at addresses specified by the document identifiers, in layers indicated by the metadata identifiers.

### 3.2.3. NLA_Doc Functions

The NLA_Doc main functions are Write and Read.

The input data are organized in a file, in CSV format. Each record contains one triple <document ID> <metadata ID> <data>. File size is not limited. The records in the file are read sequentially.

For each of them, the system does the following:

- Converts <document ID> and <metadata ID> into spatial addresses;
- Stores <data> in the point located at the address <document ID> in a layer specified by <metadata ID> in the triple.

For data retrieval, NLA_Doc uses as input a pair file

$$\textbf{<document ID>; <metadata ID>} \tag{2}$$

(each pair in a separate line) and retrieves the corresponding <data> from the archive. If some <data> do not exist, the output is empty, i.e., < > (one space).

For each of them, the system does the following:

- Converts <document ID> and <metadata ID> into spatial addresses;
- Retrieves <data> from the container located at the address <document ID> from a layer specified by <metadata ID>.

The result is a set of triplets:

$$\textbf{<document ID>; <metadata ID>; <data>} \tag{3}$$

where every triple occupies an output record.

The resulting file size is not limited. The records in the file are written sequentially.

### 3.3. Basic Measurements

The program experiments were performed on the following computer configuration:

- Processor: Intel Core2 Duo T9550 2.66 GHz; CPU Launched: 2009, Average CPU Mark: 1810 (PK = 1810);
- Physical Memory: 4.00 GB (MK = 4);
- Hard Disk: 100 GB data partition; 2 GB swap (DK = 100);
- Operating System: 64-bit operating system Windows 7 Ultimate SP1.

To perform experiments with the NLA_Doc, two datasets were prepared, containing 1000 and 10,000 instances, respectively, which are triples containing a document identifier, a metadata name and a metadata value (string): document identifiers (ten characters with letters and numbers), metadata names (six characters—word META and two numbers)

and values generated at random (Bulgarian names—character strings with variable length) (Appendix B Table A5).

The tests performed showed a recording speed of one instance from 1 to 8.5 milliseconds. The increase in time is due to the emergence of more types of metadata, the initial registration of which takes time to create the corresponding layers, and the speed varies from 100 to about 1000 instances per second (Appendix B Figure A1a).

When extracted, the speed is relatively constant and is about 1000 instances per second (Appendix B Figure A1b).

## 4. Discussion

Analyzing the results of the experiments, we can note that the main conclusions made in Reference [72] regarding NLA are valid here as well.

In all traditional relational databases, continuous reconstructions of the index structures must be done due to the incoming dynamic data. This is a major and extremely serious problem. The volume of data that is collected and indexed continuously soon becomes so large that traditional databases become overloaded and their work becomes extremely slow as they begin to use almost all the time for their own self-maintenance.

NLA-bit does not require such updates. This prevents overloading and slowing down the operation of databases, even with the accumulation of huge arrays of data. The speed of work is constant and independent of the volume of data, i.e., there is a constant time complexity $O(1)$.

In addition, important advantages of the approach are as follows:

- The reduction of the amount of occupied memory due to the complete absence of additional indexes, absolute addresses, pointers and additional files;
- Reduction of processing time due to the complete lack of demand—the data are stored/extracted to/from a direct address.

The efficiency of the NLA for storing RDF triples and quadruples was proved in Reference [72]. It was compared with such well-known system like Virtuoso, Jena and Sesame. The conclusion is that it has a very good place, showing worse time than Virtuoso but similar to Jena and better than Sesame.

In this paper, we present a possible implementation of the NLA approach for concrete example for storing document based on NLA-bit structure.

It is impossible to compare results with all existing systems, because there are no published standard benchmark data and, at the same time, it is impossible to simulate the technical base used in their concrete installations. Because of this, the value of complexity may be used for comparison.

For instance, the Neo4j uses pointers to navigate and traverse the graph. Thus, it creates additional data to support the structure and operations with them. This takes time and memory resources. At the same time, NLA-bit structures represent graphs without additional pointers and without storing NLA addresses, which permits using less memory and time.

Another well-known system, BadgerDB, is an embeddable, persistent and fast key-value (KV) database. It is the underlying database for Dgraph [79]. In addition to key-value file and indexes, structuring on the base of relations exists. This way, a two-dimensional storing structure is available. NLA-bit is a key-value structure, too. The main difference is the direct access without indexes. Thus, the complexity of BadgerDB is a least $O(\log n)$. NLA-bit has complexity $O(1)$. In addition, NL-addressing permits the use of more than two dimensions, and, in this way, it is more powerful.

The all similar systems use balanced three indexes with complexity at least $O(\log n)$. NLA approach is based on computing of hash formulas and supports direct access to dynamical perfect hash tables with complexity $O(1)$. As mental concept, NLA-bit is more powerful because, for every Subject, there exists only one NLA-bit in the world, and it is easy to establish morphisms between all NLA databases which exist or will be created in the future.

A special remark has to be made about the Create, Read, Update and Delete (CRUD) operations. NLA-bit needs only two of these operations, Read and Update; the other two are not needed because of the NLA—all points of the database space are assumed as existing but empty, and there is no need for creation and deleting. Only Update is needed to change the content and Read to receive it. Both operations have complexity O(1).

The limitations of NLA-bit are connected to available disk space on the computer or in the cloud. The main limitation is the length of NL-address, which, in different realizations, may be reduced in accordance with practical needs. There is no reason to support 1K symbols length of NL-address if used words and phrases are no longer than 100 symbols, because buffers will occupy extra memory.

The main drawback of the approach is the traditional thinking in the frame of relational model. It is difficult to jump from two- or three-dimensional space to multi-dimensional one with more than four dimensions.

## 5. Conclusions and Further Work

In this paper, a possible approach for the implementation of a distributed system for storing data for documents, related metadata and analytical results, based on NLA-bits, was presented.

A data-storing system based on NLA-bits was outlined.

The NLA-bit is a fundamentally new structure to database organization that does not replace, but naturally complements, other widely used structures of database management systems. The NLA-bits open up a wide field for research and practical implementation in the field of large databases with dynamic semi-structured data (Big Data).

This is an important direction for future work, which sets serious scientific and scientific-practical tasks. As a first next step we can point out the development of new possibilities of the presented system, which, due to the limited volume, remained out of scope of the present work. These are the activities for extracting essential data with the help of artificial intelligence functions and presentation (visualization) of summarized results to the user, which are essential parts of real automated systems. In addition, the NLA-bit structure raises new types of operations, which need corresponding functionalities of the programming languages. For instance, a possible approach may be developing of languages based on the Category theory like Haskell [80].

**Funding:** This research received no external funding.

**Institutional Review Board Statement:** Ethical review and approval are not required for this study due to artificially generation of data by a software program.

**Informed Consent Statement:** Informed consent is not required for this study due to artificially generation of data by a software program.

**Data Availability Statement:** Data available on request from the author.

**Conflicts of Interest:** The author declares no conflict of interest.

## Appendix A

**Table A1.** Thirty-two layers of NLA-bit for the document A519701/2 (only data in bold are stored in the archive; name of the document and names of the layers are NLA-addresses).

| Layer | Objects |
| --- | --- |
| URI | A519701/2 (unique registration identifier) |
| About | letter of conformance of agreement for collaboration |
| address of the recipient | Bulgaria, Sofia, 1000, PO Box 775 |
| on your № | B436213/73 |
| to № | |
| Address | |
| Application | |
| compiled | Peter Dimov |
| agreed | Damian Ivanov |
| signature | |
| resolution | Simon Nikolov |
| read | <A519701/2: read> |
| note | to be discussed on the board of directors |
| transcript | |
| type | letter of conformance |
| concerning | a brief presentation of the essence through text |
| data on the authorized person | Stefan Stanev (authorized to sign the respective type of document) |
| correspondent | name/title; address; pin/uic, etc. |
| data on the storing of the document from the departmental nomenclature: | 20 February 2020 |
| the type of file format | docx |
| attached files | path1, short description1; path2, short description2; path3, short description3; path4, short description4; . . . |
| additional comment | |
| additional data | |
| deadline for completion | 20 May 2020 |
| employee | Nikola Atanasov |
| e-signature | e-signatures of the employees who prepared and agreed on the document |
| recipient registration number | |
| method of sending/receiving | |
| creating software | |
| volume | 18 MB |
| case index | CS2468/FG83 |
| chronology | paths to descriptions of the activities related to the storing and audit of the document |

**Table A2.** Thirty-two RDF triples which correspond to NLA-bit for the document A519701/2.

| Subject | Relation | Object |
|---|---|---|
| A519701 | URI | A519701/2 (unique registration identifier) |
| A519701 | About | letter of conformance of agreement for collaboration |
| A519701 | address of the recipient | Bulgaria, Sofia, 1000, PO Box 775 |
| A519701 | on your № | B436213/73 |
| A519701 | to № | |
| A519701 | Address | |
| A519701 | Application | |
| A519701 | Compiled | Peter Dimov |
| A519701 | Agreed | Damian Ivanov |
| A519701 | Signature | |
| A519701 | Resolution | Simon Nikolov |
| A519701 | Read | <A519701/2: read> |
| A519701 | Note | to be discussed on the board of directors |
| A519701 | Transcript | |
| A519701 | Type | letter of conformance |
| A519701 | concerning | a brief presentation of the essence through text |
| A519701 | data on the authorized person | Stefan Stanev (authorized to sign the respective type of document) |
| A519701 | correspondent | name/title; address; pin/uic, etc. |
| A519701 | data on the storing of the document from the departmental nomenclature: | 20 February 2020 |
| A519701 | the type of file format | docx |
| A519701 | attached files | path1, short description1; path2, short description2; path3, short description3; path4, short description4; . . . |
| A519701 | additional comment | |
| A519701 | additional data | |
| A519701 | deadline for completion | 20 May 2020 |
| A519701 | Employee | Nikola Atanasov |
| A519701 | e-signature | e-signatures of the employees who prepared and agreed on the document |
| A519701 | recipient registration number | |
| A519701 | method of sending/receiving | |
| A519701 | creating software | |
| A519701 | volume | 18 MB |
| A519701 | case index | CS2468/FG83 |
| A519701 | chronology | paths to descriptions of the activities related to the storing and audit of the document |

**Table A3.** Six layers of NLA-bit for the subdocument <A519701/2: read> (only data in bold are stored in the archive; name of the document and names of the layers are NLA-addresses).

| Layer | Objects |
|---|---|
| 23 January 2021; 13:15:52 | Sami Mogamad Alchalian |
| 29 January 2021; 08:24:18 | Viviyan Venelinova Valkova |
| 2 February 2021; 16:38:44 | Martin Borislavov Borisov |
| 4 February 2021; 09:16:27 | Viviyan Venelinova Valkova |
| 5 February 2021; 11:52:45 | Gabriel Metodiev Krumov |
| 6 February 2021; 23:38:36 | Sami Mogamad Alchalian |

**Table A4.** Six RDF triples which correspond to NLA-bit for the subdocument <A519701/2: read>.

| Subject | Relation | Object |
|---|---|---|
| A519701: read | 23 January 2021; 13:15:52 | Sami Mogamad Alchalian |
| A519701: read | 29 January 2021; 08:24:18 | Viviyan Venelinova Valkova |
| A519701: read | 2 February 2021; 16:38:44 | Martin Borislavov Borisov |
| A519701: read | 4 February 2021; 09:16:27 | Viviyan Venelinova Valkova |
| A519701: read | 5 February 2021; 11:52:45 | Gabriel Metodiev Krumov |
| A519701: read | 6 February 2021; 23:38:36 | Sami Mogamad Alchalian |

## Appendix B

**Table A5.** Sample instances of input (a), test (b) and extracted data (c).

A519701/2; META62; NIKOLA BRANIMIROV IVANOV
B436213/73; META82; GRIGOR GAVRILOV GRANDZHEV
C531719/36; META68; LACHEZAR VENTSISLAVOV IVANOV
D448266/7; META43; SAMI MOGAMAD ALCHALIAN
E847955/73; META38; VIVIYAN VENELINOVA VALKOVA
F326742/15; META32; MARTIN BORISLAVOV BORISOV
G356217/15; META50; GABRIEL METODIEV KRUMOV
H430291/27; META71; ANGEL ALEKSANDROV BOYADZHIEV
I239507/17; META95; STANISLAV SVETLOZAROV ZLATEV
J31634/40; META39; MARTIN MITKOV DIMITROV
. . .
(a) sample Input data
A519701/2; META62;
B436213/73, META82;
C531719/36, META68;
D448266/7, META43;
E847955/73, META38;
F326742/15, META32;
G356217/15; META50;
H430291/27, META71;
I239507/17; META95;
J31634/40, META39;
. . .

**Table A5.** *Cont.*

(b) sample test data
A519701/2; META62; NIKOLA BRANIMIROV IVANOV;
B436213/73; META82; GRIGOR GAVRILOV GRANDZHEV;
C531719/36; META68; LACHEZAR VENTSISLAVOV IVANOV;
D448266/7; META43; SAMI MOGAMAD ALCHALIAN;
E847955/73; META38; VIVIAN VENELINOVA VALKOVA;
F326742/15; META32; MARTIN BORISLAVOV BORISOV;
G356217/15; META50; GABRIEL METODIEV KRUMOV;
H430291/27; META71; ANGEL ALEXANDROV BOYADZHIEV;
I239507/17; META95; STANISLAV SVETLOZAROV ZLATEV;
J31634/40; META39; MARTIN MITKOV DIMITROV;
. . .
(c) sample extracted data

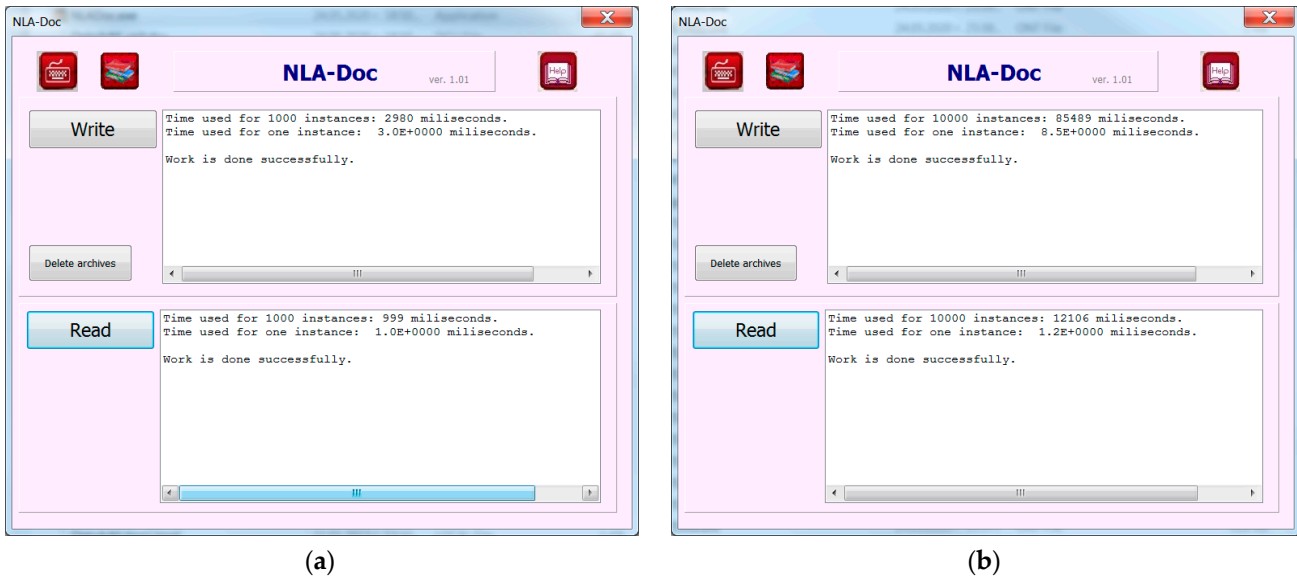

(**a**)          (**b**)

**Figure A1.** Results for recording and retrieving (**a**) 1000 documents and (**b**) 10,000 documents.

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
