# Peer review of "NLA-Bit: A Basic Structure for Storing Big Data with Complexity O(1)"

_2504-2289, doi:10.3390/bdcc5010008_

Round 1

Reviewer 1 Report

The paper presents a data storing structure called NLA_bit, which serves as a storing structure for semantic data in the form of RDF triples. The proposed method builds upon NLA storing systems and extends it with the addition of relationship naming (the metadata). The description and formulation of the proposed method are satisfactory. Still, there is a lack of any real literature review on the possibilities of storing RDF efficiently (there is no mention of any of the graph databases). The paper's major issue is the experimental part, as there is no real experiment for the validation of the proposed method. The experiment lack testing of common RDF properties (mention later on) and any meaningful comparison to existing systems. The paper's structure is appropriate, but the paper should be proofread by a native English speaker, as there are grammar mistakes in the paper.

-The introduction only bearly mentions the novelty "NLA_bit." There are no apparent contributions, which are an integral part of every introduction. Please state what does NLA-bit contributes to the already existing body of knowledge on this scientific field. Comparing both works, the main difference is the storing of the domain with NLA_bit. Still, this should be already apparent after reading the introduction.

-The differences between already published research from the same author, "_Natural Language Addressing" in 2005, and this paper are not apparent. The phrase NLA_bit is new, but the program and the experiment were already published in that research.

-In RDF, one subject can have multiple connections/relations of the same type of object. How is this addressed in NLA_bit? As the subject and the relation are two identifiers leading to one information, there is no obvious solution for this common occurrence. Let's say you have a document addressed to multiple people in one organization - this would be presented with numerous "addressed_to" relations to multiple different objects. In your implementation, the combination of the subject and the relation lead to just one content. Would the content be separated somehow? If yes, then you do not have O(1) complexity anymore.

-Furthermore, how would you present multiple relationships between two of the same subject and objects. Let's say we have a document made (first relation) by one author and was also sent to the same author (second relation). With NLA_link, this would have to be duplicated. If this would be the case, how would you represent that the duplicated objects are the same, which is a crucial RDF feature? Would a new relation be added, which would serve to equalize several contents? How would this impact the efficiency of the proposed approach?

-The experiment lacks any meaningful comparison to competing systems. You could use any of the graph databases (i.e., Neo4j) to validate your systems. If your hypothesis is correct, your system should cover allč of the edge cases and be faster. If edge cases are not covered, then NLA_bit is not appropriate for RDF storing.

-The proposed systems should be tested in all CRUD operations. It is not needed that your system is better in all (or any) of the operations, but any new proposal needs a benchmark point so that the readers can see if the proposed method is useful for their problem.

Author Response

Dear Reviewer,

Thank you very much for the fruitful remarks. Please see below my solutions.

Respectfully yours

Author

Remark 1:

The paper presents a data storing structure called NLA_bit, which serves as a storing structure for semantic data in the form of RDF triples. The proposed method builds upon NLA storing systems and extends it with the addition of relationship naming (the metadata). The description and formulation of the proposed method are satisfactory. Still, there is a lack of any real literature review on the possibilities of storing RDF efficiently (there is no mention of any of the graph databases). The paper's major issue is the experimental part, as there is no real experiment for the validation of the proposed method. The experiment lack testing of common RDF properties (mention later on) and any meaningful comparison to existing systems. The paper's structure is appropriate, but the paper should be proofread by a native English speaker, as there are grammar mistakes in the paper.

Solution: rows 39-58

1.1. Graph databases

Many of the semi-structured models are "graph-like". This brought attention back to the graph models. These are models in which the data structures for the schema and / or instances are represented as a directed, probably labeled, graph. Data manipulation is expressed through graph-oriented operations [5].

Database graph models emerged in the 1980s and early 1990s, and their influence gradually diminished with the advent of other database models, in particular geographic, spatial, semi-structured, and XML models. Graph databases are used in areas where information about the interconnectedness or topology of the data is more important than or as important as the data itself. Graph databases use a model in which the data structures for the schema and / or instances are represented as an oriented graph, and the data manipulation is set by graph-oriented operations [5].

We may classify the graph data bases’ models in two main classes:

  • With explicit schema, such as: Logical Data Model (LDM) [68, 69]; Hypernode Mode (HyM); [71; 80; 70]; GROOVY [72]; GOOD [60; 54]. GMOD [46]; PaMaL [54]; GOAL [64]; GDM [65, 66]; Gram [45]. The explicit schema exists separately from the data in database and is used as an external pattern for data organization.
  • With implicit schema such as: Object Exchange Model (OEM) [79]; GGL [55; 56; 57; 58]; Simatic-XT [73]; RDF [6; 47; 62]. Implicit schema is integrated in the data base as a root of the data structure. In other words, the first data structure which can be accessed is the schema from which all other structures may be traversed using it as a pattern.

rows 98-170

1.3. Technologies for storing RDF data

Good reviews of existing technologies for storing and retrieving RDF data are given in [63] and [1]. Technologies for storing RDF data can be classified into:

(1) General schemes, i.e. schemes that do not support certain structures and run on third-party databases, such as Jena SDB, which can connect to almost all relational databases such as MySQL, PostsgreSQL and Oracle;

(2) Specific to schemas that provide storing with their own database structures (Virtuoso, Mulgara, AllegroGraph and Garlik JXT).

1.3.1. General schemes

1.3.1.1. Vertical presentation

This is the simplest RDF schema with only one table needed in the database.

The table contains three columns called Subject, Relation and Object. Indexes are added for each column. [44]. The biggest advantage of this scheme is that it does not require restructuring in case of data changes. This approach is used by Oracle [77; 51], 3store [24], Redland [32], RDFStore [31] and rdfDB [59]. In many publications “Relation” is named as a “Property” with the same meaning.

1.3.1.2. Normalized triple store

Additional idea is the normalized triple store. It consists of adding two additional tables to store URI and literals separately, which requires significantly less storage space [24] (Table 1). A hybrid of simple and normalized triple storage is also possible, in which the storing of the values themselves is either in the triple table or in the resource table [15].

Table 1. Normalized triple store

Triples:

Resources:

Literals:

Subject

Relation

IsLiteral

Object

r1

r2

False

r3

r1

r4

True

l1

ID

URI

r1

…#1

r2

…#2

ID

Value

l1

Value1

The table can be divided horizontally into several tables, each of which represents a separate relation. These tables only need two columns for Subject and Object. Table names implicitly contain relations [50; 53].

To implement the vertical partitioning approach, the tables must be stored using a column-oriented DBMS (i.e. a DBMS designed specifically for this case. The columns can be indexed (e.g. using a non-clustered B + tree). The main advantage of vertical partitioning is the support for fast object join. This is achieved by sorting tables. The vertical partition approach offers support for multi valued attributes. In fact, if an object has more than one object value for a given property, each individual value is listed sequentially in the table for this property. For a query, only the relations included in that query need to be read, and no clustering algorithm is required to split the table of triplets into tables with two columns. Inserts are slow because multiple tables need to be accessed. [1].

1.3.2. Specific schemas

1.3.2.1. Horizontal presentation

The basic schema consists of a table with a column for the instance identifier (ID), a column for the class name, and a column for each relation. Thus, one row in the table corresponds to one instance. This scheme corresponds to the horizontal presentation [44] and obviously has several drawbacks: Large number of columns; High sparsity; Inability to handle multi-valued properties; The need to add columns to the table when adding new properties to the ontology, etc.

1.3.2.2. Vertical decomposition

The vertical decomposition results in one relation table with only two columns for Subject and Object. It's called a decomposition storage model. Relationships are also stored in tables, e.g. the rdf: type table contains the relationships between instances and their classes [63]. A similar hybrid scheme is used to use the combination of both class table and class table relation schemes. (Table 2) [78].

Table 2. RDF Hybrid schema (the table-per-relation approach)

ClassA:

Relation1:

ClassB:

ID

…#1

Subject

Object

…#1

…#3

ID

…#3

The main disadvantage of this approach is the generation of many NULL values, as not all relations will be defined for all subjects. In addition, the ambiguous attributes that are common in RDF data are difficult to express. When searching for all defined relations of an object, scanning of all tables is required. The inclusion of new relationships also requires the addition of new tables [1]. This approach has been used by systems such as Sesame [18; 49], Jena2 [15; 16], RDFSuite [20] and 4store [25].

1.3.2.3. Multi-indexing

The idea of multi-indexing is based on the fact that relationship-related queries are not necessarily the most interesting or popular type of queries found in the real-world semantic web applications.

RDF data should be processed equally by the following type of requests:

  • triples with the same subject;
  • triples with equal relations;
  • list of subjects or relations related to an object.

To this end, a set of six indexes is maintained, covering all possible access schemes that an RDF request may require. These indices are RSO, ROS, SRO, SOR, ORS and OSR (R means relation, O for object and S for subject).

At first glance, such multiple indexing would lead to a combinatorial explosion for a simple relational table. Nevertheless, this is quite practical in the case of RDF data [8, 14]. The approach pays equal attention to all elements of RDF [1]. This approach has been used by tools such as the Kowari system [26], Virtuoso [29], RDF-3X [28], Hexastore [14], RDFCube [35], BitMat [13], BRAHMS [67], RDFJoin [74], RDFKB [75], TripleT [52], iStore [81], Parliament [34].

rows 172-173

The analysis of the considered tools showed that they all use one or more indexes for access to the data. So, their complexity is at least O(log n).

Remark 2:

-The introduction only bearly mentions the novelty "NLA_bit." There are no apparent contributions, which are an integral part of every introduction. Please state what does NLA-bit contributes to the already existing body of knowledge on this scientific field. Comparing both works, the main difference is the storing of the domain with NLA_bit. Still, this should be already apparent after reading the introduction.

Solution:

rows 171-204

1.4. The goal of this paper

The analysis of the considered tools showed that they all use one or more indexes for access to the data. So, their complexity is at least O(log n).

In this paper, we introduce a novel approach for storing RDF data based on the possibilities of Natural Language Addressing (NLA) [36], [43] and on a special NLA basic structure for storing Big Data, called “NLA-bit”, which is aimed to support very large distributed RDF triple or quadruple stores with time complexity O(1).

In all traditional relational databases, continuous reconstructions of the index structures must be done due to the incoming dynamic data. This is a major and extremely serious problem. The volume of data that is collected and indexed continuously soon becomes so large that traditional databases become overloaded and their work becomes extremely slow as they begin to use almost all the time for their own self-maintenance.

NLA_bit does not require such updates. This prevents overloading and slowing down the operation of databases, even with the accumulation of huge arrays of data. The speed of work is constant and independent of the volume of data, i.e. there is a constant time complexity O(1).

In addition, important advantages of the approach are:

  • The reduction of the amount of occupied memory due to the complete absence of additional indexes, absolute addresses, pointers, and additional files;
  • Reduction of processing time due to the complete lack of demand - the data is stored/extracted to/from a direct address.

The NLA_bit is a fundamentally new structure to database organization that does not replace, but naturally complements, other widely used structures of database management systems.

Applying the idea of NLA to the dynamical perfect hash tables supported by MDNDB® [40], it is possible to realize the same approach as multi-indexing without indexes but with direct access to all data elements. In addition, because of not storing keywords used as addresses, the solution will occupy les memory.

So, the structure NLA-bit, introduced in this paper may be used in three variants - for subjects, for relations, as well as for objects. Below we will outline only the case for subjects.

Presented in this paper research continues the work on possibilities of Natural Language Addressing presented in 2015 year in [36] in direction to establish special structures suitable for working just with RDF instances. It is important that the instances may be triples, quadruples, as well as with many more elements. Proving the possibilities of dynamical perfect hash tables for NLA was done in [36]. In this work, the research continues with defining a new structure which may be useful for building RDF stores.

Remark 3:

-The differences between already published research from the same author, "_Natural Language Addressing" in 2005, and this paper are not apparent. The phrase NLA_bit is new, but the program and the experiment were already published in that research.

Solution:

rows 199-204

Presented in this paper research continues the work on possibilities of Natural Language Addressing presented in 2015 year in [36] in direction to establish special structures suitable for working just with RDF instances. It is important that the instances may be triples, quadruples, as well as with many more elements. Proving the possibilities of dynamical perfect hash tables for NLA was done in [36]. In this work, the research continues with defining a new structure which may be useful for building RDF stores.

Remark 4:

-In RDF, one subject can have multiple connections/relations of the same type of object. How is this addressed in NLA_bit? As the subject and the relation are two identifiers leading to one information, there is no obvious solution for this common occurrence. Let's say you have a document addressed to multiple people in one organization - this would be presented with numerous "addressed_to" relations to multiple different objects. In your implementation, the combination of the subject and the relation lead to just one content. Would the content be separated somehow? If yes, then you do not have O(1) complexity anymore.

Solution:

rows 263-269

If the practical solution needs processing of requests based on NLA-bits for relations or NLA-bits for objects, they may be used in parallel with NLA-bits for subjects. If the Subject is connected with multiple relations they are separate addresses in the NLA-bit space and no conflicts exist. If a Subject with concrete Relation is connected to multiple Objects, the solution may be trivial to apply classical concatenation of all objects in one address point, or specific for NLA to use additional RDF elements such as context to address separated different values of the Objects. In the last case the complexity is still O(1).

Remark 5:

-Furthermore, how would you present multiple relationships between two of the same subject and objects. Let's say we have a document made (first relation) by one author and was also sent to the same author (second relation). With NLA_link, this would have to be duplicated. If this would be the case, how would you represent that the duplicated objects are the same, which is a crucial RDF feature? Would a new relation be added, which would serve to equalize several contents? How would this impact the efficiency of the proposed approach?

Solution:

rows 270-275

Interesting case is when the Subject and Object are connected by multiple Relations. One possible realization of such case is to use NLA-bits for relations. Other variant is to use separate points for all combinations of Subject and Relations which will store object many times in all corresponded points. At the end, the most efficient variant is to use Subject and Object as NLA addresses and to store all relations in corresponded points using additional NLA addresses. In all variants, the complexity is O(1).

Remark 6:

-The experiment lacks any meaningful comparison to competing systems. You could use any of the graph databases (i.e., Neo4j) to validate your systems. If your hypothesis is correct, your system should cover allč of the edge cases and be faster. If edge cases are not covered, then NLA_bit is not appropriate for RDF storing.

Solution:

rows 508-531

The efficiency of the NLA for storing RDF triples and quadruples was proved in [36]. It was compared with such well known system like Virtuoso, Jena and Sesame. The conclusion is that it has very good place showing worst time than Virtuoso but similar to Jena and better than Sesame. This result is visualized in [36] by the Nemenyi test [45].

In this paper, we present a possible implementation of the NLA approach for concrete example for storing document based on NLA-bit structure.

It is impossible to compare results with all existing systems because there are no published standard benchmark data and in the same time it is impossible to simulate the technical base used in their concrete installations. Because of this, the value of complexity may be used for comparison.

For instance, the Neo4j uses pointers to navigate and traverse the graph. So it creates additional data to support the structure and operations whit it. This takes time and memory resources. In the same time NLA-bit structures represent graphs without additional pointers and without storing NLA addresses which permits using less memory and time.

Another well known system, BadgerDB, is an embeddable, persistent, and fast key-value (KV) database. It is the underlying database for Dgraph [48]. In addition to key-value file and indexes, structuring on the base of relations exists. This way a two dimensional storing structure is available. NLA-bit is a key-value structure, too. The main difference is the direct access without indexes. So, the complexity of BadgerDB is a least O(log n). NLA-bit has complexity O(1). In addition, NL-addressing permits using more than two dimensions and this way it is more powerful.

The all similar systems use balanced three indexes with complexity at least O(log n). NLA approach is based on computing of hash formulas and support direct access to dynamical perfect hash tables with complexity O(1). As mental concept, NLA-bit is more powerful because for every subject there exist only one NLA-bit in the world and it is easy to establish morphisms between all NLA data bases which exist or will be created in the future.

Remark 7:

-The proposed systems should be tested in all CRUD operations. It is not needed that your system is better in all (or any) of the operations, but any new proposal needs a benchmark point so that the readers can see if the proposed method is useful for their problem.

Solution:

rows 532-536

A special remark has to be done about the CRUD operations. NLA-bit needs only two of these operations Read and Update, the other two are not needed because of the NLA - all points of the data base space are assumed as existing but empty and there is no need of creation and deleting. Only Update is needed to change the content and Read to receive it. Both operations have complexity O(1).

Reviewer 2 Report

This paper introduces a novel method of storing RDF triples based on Natural Language Addressing for the purpose of reducing time complexity of data access. The presented method is quite interesting but the presentation of this paper is poor from a scientific perspective. 

First, there is no clear evaluation result to show that the presented method is better than any existing approach. Without any quantitative measurement results for this type comparison, it is difficult to convince the readers about the novelty of this new approach.

Second, the explanation of “NLA_domain” is weak and unclear.

Third, the Result section is a mixture of different aspects: use cases, system design, and some basic measurements. It is good to separate them out to have a clearer structure.

Last, this paper is lack of the analysis about the limitation and drawback of this approach. For example, as compared to the other approaches, like the one in DGraph, which distributes the storage based on relations, what is the pros and cons of the presented new approach?

Author Response

Dear Reviewer,

Thank you very much for the fruitful remarks. Please see below my solutions.

Respectfully yours

Author

Remark 1:

This paper introduces a novel method of storing RDF triples based on Natural Language Addressing for the purpose of reducing time complexity of data access. The presented method is quite interesting but the presentation of this paper is poor from a scientific perspective.

Solution:

The paper was edited to answer to this remark.

Remark 2:

First, there is no clear evaluation result to show that the presented method is better than any existing approach. Without any quantitative measurement results for this type comparison, it is difficult to convince the readers about the novelty of this new approach.

Solution:

rows 508-536

The efficiency of the NLA for storing RDF triples and quadruples was proved in [36]. It was compared with such well known system like Virtuoso, Jena and Sesame. The conclusion is that it has very good place showing worst time than Virtuoso but similar to Jena and better than Sesame. This result is visualized in [36] by the Nemenyi test [45].

In this paper, we present a possible implementation of the NLA approach for concrete example for storing document based on NLA-bit structure.

It is impossible to compare results with all existing systems because there are no published standard benchmark data and in the same time it is impossible to simulate the technical base used in their concrete installations. Because of this, the value of complexity may be used for comparison.

For instance, the Neo4j uses pointers to navigate and traverse the graph. So it creates additional data to support the structure and operations whit it. This takes time and memory resources. In the same time NLA-bit structures represent graphs without additional pointers and without storing NLA addresses which permits using less memory and time.

Another well known system, BadgerDB, is an embeddable, persistent, and fast key-value (KV) database. It is the underlying database for Dgraph [48]. In addition to key-value file and indexes, structuring on the base of relations exists. This way a two dimensional storing structure is available. NLA-bit is a key-value structure, too. The main difference is the direct access without indexes. So, the complexity of BadgerDB is a least O(log n). NLA-bit has complexity O(1). In addition, NL-addressing permits using more than two dimensions and this way it is more powerful.

The all similar systems use balanced three indexes with complexity at least O(log n). NLA approach is based on computing of hash formulas and support direct access to dynamical perfect hash tables with complexity O(1). As mental concept, NLA-bit is more powerful because for every subject there exist only one NLA-bit in the world and it is easy to establish morphisms between all NLA data bases which exist or will be created in the future.

A special remark has to be done about the CRUD operations. NLA-bit needs only two of these operations Read and Update, the other two are not needed because of the NLA - all points of the data base space are assumed as existing but empty and there is no need of creation and deleting. Only Update is needed to change the content and Read to receive it. Both operations have complexity O(1).

Remark 3:

Second, the explanation of “NLA_domain” is weak and unclear.

Solution:

It is deleted from the text. I plan to explain it in other paper.

Remark 4:

Third, the Result section is a mixture of different aspects: use cases, system design, and some basic measurements. It is good to separate them out to have a clearer structure.

Solution:

It is done by new structuring the chapter.

Remark 5:

Last, this paper is lack of the analysis about the limitation and drawback of this approach. For example, as compared to the other approaches, like the one in DGraph, which distributes the storage based on relations, what is the pros and cons of the presented new approach?

Solution:

rows 514-544

It is impossible to compare results with all existing systems because there are no published standard benchmark data and in the same time it is impossible to simulate the technical base used in their concrete installations. Because of this, the value of complexity may be used for comparison.

For instance, the Neo4j uses pointers to navigate and traverse the graph. So it creates additional data to support the structure and operations whit it. This takes time and memory resources. In the same time NLA-bit structures represent graphs without additional pointers and without storing NLA addresses which permits using less memory and time.

Another well known system, BadgerDB, is an embeddable, persistent, and fast key-value (KV) database. It is the underlying database for Dgraph [48]. In addition to key-value file and indexes, structuring on the base of relations exists. This way a two dimensional storing structure is available. NLA-bit is a key-value structure, too. The main difference is the direct access without indexes. So, the complexity of BadgerDB is a least O(log n). NLA-bit has complexity O(1). In addition, NL-addressing permits using more than two dimensions and this way it is more powerful.

The all similar systems use balanced three indexes with complexity at least O(log n). NLA approach is based on computing of hash formulas and support direct access to dynamical perfect hash tables with complexity O(1). As mental concept, NLA-bit is more powerful because for every subject there exist only one NLA-bit in the world and it is easy to establish morphisms between all NLA data bases which exist or will be created in the future.

A special remark has to be done about the CRUD operations. NLA-bit needs only two of these operations Read and Update, the other two are not needed because of the NLA - all points of the data base space are assumed as existing but empty and there is no need of creation and deleting. Only Update is needed to change the content and Read to receive it. Both operations have complexity O(1).

The limitations of NLA-bit are connected to available disk space on the computer or in the cloud. The main limitation is the length of NL-address, which in different realizations may be reduced in accordance with practical needs. There is no reason to support 1K symbols length of NL-address if used words and phrases are no longer than 100 symbols because buffers will occupy extra memory.

The main drawback of the approach is the traditional thinking in the frame of relational model. It is difficult to jump from two or three dimensional space to multi-dimensional one with more than 4 dimensions.

Reviewer 3 Report

Author proposes an approach for the implementation of a distributed system for storing data for documents and related metadata and analytical results, based on NLA_bits.

The paper describes an interesting topic that is within the scope of the journal. It is well written. The proposal is original. It can be read fluently. It also has an adequate number of references.

With regard to content, it presents results of significant impact assuming progress in this domain.

The main improvement is to extend the discussion section to evaluate the proposal with other similar proposals. In this sense, it would be good to have a comparative table showing the differences and the advantages and disadvantages of each approach.

Also, it would be wise not to use so many web page references and try to use journal articles. It would also be interesting to include a section on conclusions and future work. In particular, indicate how the proposed work could be continued and its possible applications.

Author Response

Dear Reviewer,

Thank you very much for the fruitful remarks. Please see below my solutions.

Respectfully yours

Author

Author proposes an approach for the implementation of a distributed system for storing data for documents and related metadata and analytical results, based on NLA_bits.

The paper describes an interesting topic that is within the scope of the journal. It is well written. The proposal is original. It can be read fluently. It also has an adequate number of references.

With regard to content, it presents results of significant impact assuming progress in this domain.

Remark 1:

The main improvement is to extend the discussion section to evaluate the proposal with other similar proposals. In this sense, it would be good to have a comparative table showing the differences and the advantages and disadvantages of each approach.

Solution:

rows 508-544

The efficiency of the NLA for storing RDF triples and quadruples was proved in [36]. It was compared with such well known system like Virtuoso, Jena and Sesame. The conclusion is that it has very good place showing worst time than Virtuoso but similar to Jena and better than Sesame. This result is visualized in [36] by the Nemenyi test [45].

In this paper, we present a possible implementation of the NLA approach for concrete example for storing document based on NLA-bit structure.

It is impossible to compare results with all existing systems because there are no published standard benchmark data and in the same time it is impossible to simulate the technical base used in their concrete installations. Because of this, the value of complexity may be used for comparison.

For instance, the Neo4j uses pointers to navigate and traverse the graph. So it creates additional data to support the structure and operations whit it. This takes time and memory resources. In the same time NLA-bit structures represent graphs without additional pointers and without storing NLA addresses which permits using less memory and time.

Another well known system, BadgerDB, is an embeddable, persistent, and fast key-value (KV) database. It is the underlying database for Dgraph [48]. In addition to key-value file and indexes, structuring on the base of relations exists. This way a two dimensional storing structure is available. NLA-bit is a key-value structure, too. The main difference is the direct access without indexes. So, the complexity of BadgerDB is a least O(log n). NLA-bit has complexity O(1). In addition, NL-addressing permits using more than two dimensions and this way it is more powerful.

The all similar systems use balanced three indexes with complexity at least O(log n). NLA approach is based on computing of hash formulas and support direct access to dynamical perfect hash tables with complexity O(1). As mental concept, NLA-bit is more powerful because for every subject there exist only one NLA-bit in the world and it is easy to establish morphisms between all NLA data bases which exist or will be created in the future.

A special remark has to be done about the CRUD operations. NLA-bit needs only two of these operations Read and Update, the other two are not needed because of the NLA - all points of the data base space are assumed as existing but empty and there is no need of creation and deleting. Only Update is needed to change the content and Read to receive it. Both operations have complexity O(1).

The limitations of NLA-bit are connected to available disk space on the computer or in the cloud. The main limitation is the length of NL-address, which in different realizations may be reduced in accordance with practical needs. There is no reason to support 1K symbols length of NL-address if used words and phrases are no longer than 100 symbols because buffers will occupy extra memory.

The main drawback of the approach is the traditional thinking in the frame of relational model. It is difficult to jump from two or three dimensional space to multi-dimensional one with more than 4 dimensions.

Remark 2:

Also, it would be wise not to use so many web page references and try to use journal articles. It would also be interesting to include a section on conclusions and future work. In particular, indicate how the proposed work could be continued and its possible applications.

Solution:

The web page reverences for journal articles and books were removed from the bibliography and 38 new titles were added.

Conclusion and future work section was added.

rows 545-558

  1. Conclusion and further work

In this paper, a possible approach for the implementation of a distributed system for storing data for documents and related metadata and analytical results, based on NLA_bits, was presented.

A data storing system based on NLA_bits was outlined.

The NLA_bit is a fundamentally new structure to database organization that does not replace, but naturally complements, other widely used structures of database management systems. The NLA_bits open up a wide field for research and practical implementation in the field of large databases with dynamic semi-structured data (Big Data).

This is an important direction for future work, which sets serious scientific and scientific-practical tasks. As a first next step we can point out the development of new possibilities of the presented system, which due to the limited volume remained out of scope of the present work. These are the activities for extracting essential data with the help of artificial intelligence functions and presentation (visualization) of summarized results to the user, which are essential parts of real automated systems.

Round 2

Reviewer 1 Report

Thank you for the response. The author improved the paper in accordance with some of the issues I and other reviewers raised. Still, some of the answers and proposed solutions to the raised issues are unsatisfactory, and thus further justifications are necessary if this paper should be considered for publication.

-Let's touch on the problem of representing multiple relations between the same pair of a subject and an object. The authors responded that different relations should be represented with numerous NLA_bits but do not explain how this would be done. The issue is swept aside in one sentence, without any real solution, besides "use NLA_bits for relations."

-The answers to my issues and the other reviewers' issues about the NLA_bits problems are answered with that it has already been shown in the reference [36] or [45]. Looking at those references, it is evident that the references are problematic. The [36] reference is in a journal, where the authors of [36] are also on the editorial board - even the Editor in Chief! That alone shows the conflict of interest and puts the paper into question.

-The author is misleading by referencing the paper [45] as proof with the Nemenyi test. That paper does not include any statistical tet whatsoever.

-It is wrong stating, that the computational complexity is O(1) when there are multiple similar NLA_bits present (i.e., in case of same relations). If you feel that it is correct, please provide further justification.

Author Response

Dear reviewer,

Thank you for additional remarks. Please find  my answers in the attachment.

Respectfully yours,

Author

Reviewer 2 Report

Thanks for the answers.

The added content does provide some useful information and addressed the previous concerns. Although the current shape of this paper is not perfect, but it does give some values and insights to the readers about this new method called "NLA-bit". 

Please check the small typo-error in the updated part, e.g., "NLA-bitNLA-bit" 

Author Response

Dear reviewer,

Thank you for additional remark. Please find below my answer.

Thanks for the answers.

The added content does provide some useful information and addressed the previous concerns. Although the current shape of this paper is not perfect, but it does give some values and insights to the readers about this new method called "NLA-bit". 

Remark 1:

Please check the small typo-error in the updated part, e.g., "NLA-bitNLA-bit" 

Solution:

The typo-errors were removed.

Respectfully yours

Author
